# Unite and Conquer: Association of Two G-Quadruplex Aptamers Provides Antiproliferative and Antimigration Activity for Cells from High-Grade Glioma Patients

**DOI:** 10.3390/ph17111435

**Published:** 2024-10-26

**Authors:** Svetlana Pavlova, Lika Fab, Fatima Dzarieva, Anastasia Ryabova, Alexander Revishchin, Dmitriy Panteleev, Olga Antipova, Dmitry Usachev, Alexey Kopylov, Galina Pavlova

**Affiliations:** 1Institute of Higher Nervous Activity and Neurophysiology, Russian Academy of Sciences, 117485 Moscow, Russia; 2Institution N. N. Burdenko National Medical Research Center of Neurosurgery of the Ministry of Health of the Russian Federation, 125047 Moscow, Russia; 3Prokhorov General Physics Institute of the Russian Academy of Sciences, 119991 Moscow, Russia; 4Belozersky Research Institute of Physical Chemical Biology, Lomonosov Moscow State University, 119991 Moscow, Russia

**Keywords:** high-grade glioma, cell migration, aptamers, G-quadruplexes, anti-proliferative activity, anti-migration activity, treatment

## Abstract

**Background**: High-grade gliomas remain a virtually incurable form of brain cancer. Current therapies are unable to completely eradicate the tumor, and the tumor cells that survive chemotherapy or radiation therapy often become more aggressive and resistant to further treatment, leading to inevitable relapses. While the antiproliferative effects of new therapeutic molecules are typically the primary focus of research, less attention is given to their influence on tumor cell migratory activity, which can play a significant role in recurrence. A potential solution may lie in the synergistic effects of multiple drugs on the tumor. **Objectives**: In this study, we investigated the effect of combined exposure to bi-(AID-1-T), an anti-proliferative aptamer, and its analog bi-(AID-1-C), on the migratory activity of human GBM cells. **Results**: We examined the effects of various sequences of adding bi-(AID-1-T) and bi-(AID-1-C) on five human GBM cell cultures. Our findings indicate that certain sequences significantly reduced the ability of tumor cells to migrate and proliferate. Additionally, the expression of Nestin, PARP1, L1CAM, Caveolin-1, and c-Myc was downregulated in human GBM cells that survived exposure, suggesting that the treatment had a persistent antitumor effect on these cells.

## 1. Introduction

Oncology is one of the most devastating categories of human pathologies. Various types of cancer have been studied by specialists to varying degrees, and many of them have identified therapeutic approaches. However, brain tumors continue to present exceptional treatment challenges. Given that the brain functions as the center of guidance for all human activity, medicine encounters unique challenges due to its uniqueness and irreplaceability. Unlike paired organs, such as the kidneys, the brain cannot be excised, and any surgical intervention involving it is highly invasive and traumatic due to its complex anatomy.

Glioma is a neuroectodermal tumor primarily located in the brain, although it can also occur in the spinal cord. According to the World Health Organization (WHO) 2021 classification [1], gliomas are divided into four grades of malignancy, ranging from I to IV, with Grades III and IV being the most aggressive. Patients diagnosed with Grade IV glioma, known as glioblastoma multiforme (GBM), have the lowest survival rates, averaging approximately one year [1,2,3]. Management of high-grade gliomas remains a complex challenge for contemporary science and medicine. The current approaches and strategies used today only serve to delay disease progression rather than provide a cure. A significant factor contributing to this issue is the resistance of tumor cells to radiotherapy and chemotherapy. The primary cause of recurrence is the high rate of proliferation and migration of tumor cells. Numerous studies [4,5,6,7] have demonstrated that therapeutic interventions, including surgical excision, chemotherapy, and radiotherapy, can induce similar invasive properties in the surviving tumor cells. This phenomenon leaves patients vulnerable to disease recurrence, often with more therapy-resistant characteristics. Furthermore, it is known that tumor cells can infiltrate healthy brain tissue both spontaneously and in response to treatment, leading to recurrences [8,9]. The limitations of existing diagnostic techniques in detecting these cells make cell removal efforts unfeasible.

Although targeting survived cells could significantly enhance treatment outcomes, the currently available techniques and the specific location of the tumor impose a number of significant limitations. For instance, radiotherapy and chemotherapy can have detrimental effects on the patient’s health, leading to numerous complications and potentially causing the tumor to develop resistance and become more aggressive. Surgical treatment often fails to completely remove the tumor, and additional therapies are ineffective [4,10,11] It has been shown, for example, that temozolomide (TMZ), the most commonly used chemotherapeutic agent for treating GBM, does not effectively target actively migrating cells; in fact, it may promote increased invasion of surviving GBM cells [7]. A similar effect is observed with radiation therapy, which can effectively stop tumor growth, but the surviving cells tend to become more migratory, resulting in subsequent relapses [6,12,13].

Current research is actively exploring the possibility of using antibodies to treat tumors [14,15,16]. Antibodies are proteins that can specifically bind to target proteins and deliver therapeutic agents to cells in a targeted manner. However, they have several disadvantages, including high molecular weight, low stability, significant variability, challenges in production, high costs, and elevated immunogenicity [16]. Furthermore, tumor cells that survive antibody treatment often exhibit increased migratory properties; this has been shown, for instance, following the use of bevacizumab (an anti-VEGF IgG1 monoclonal antibody) [5]. These factors have led to the necessity for the exploration of novel therapeutic modalities.

Aptamers have become one of the most recent areas of research. These short oligonucleotides are capable of binding to a wide range of targets. It is a single-stranded RNA or DNA oligonucleotides that fold into a three-dimensional structure, enabling them to recognize specific targets based on the “key to lock” principle [17]. They are characterized by their small size, stability, and ease of synthesis. Because they are chemically synthesized, their composition and purity can be precisely controlled, and they are non-immunogenic [18,19,20]. Due to their small size (5–50 kDa), aptamers can freely penetrate cells and traverse the blood–brain barrier [21,22]. However, these properties are not universally applicable to all aptamers, necessitating further research to fully understand the characteristics of each aptamer. Because aptamers bind to certain target proteins with high specificity and affinity, they can be considered super-small analogs of antibodies. Furthermore, aptamers mostly exhibit a cytostatic effect in tumor therapy, which helps to spare healthy cells from damage. Nevertheless, they also have disadvantages that hinder their development into fully-fledged drugs. One significant drawback is the instability of the molecules, which leads to a rapid decline in their therapeutic effects [17,22]. Additionally, research has shown that using aptamers to inhibit cell proliferation can result in increased migratory activity among the surviving cells, a highly detrimental outcome for disease progression [23,24]. Moreover, this effect is typical not only of aptamers but also observed with other effective therapeutic agents [5,7,25]. In this context, the exploration of synergistic effects are attracting more and more attention [26,27,28,29].

This approach may prove to be significantly more effective, as each molecule will “attack” its specific target, thereby enhancing therapeutic exposure. Interest in combination therapies is becoming popular for traditional approaches. For instance, phase II clinical trials have demonstrated the efficacy of combining standard therapy with tislelizumab and low-dose bevacizumab, which has shown effectiveness in 32 patients with glioblastoma (GBM) who experienced their first recurrence after standard therapy [30]. Another study using a combination therapy has shown that the combined effect of progesterone (Prog) and abiraterone (Abi) can substantially improve the efficacy of GBM treatment by modulating the tumor microenvironment and enhancing natural killer (NK) cell-mediated immunity [31]. A combination regimen consisting of sintilimab, bevacizumab, and TMZ is also being considered for the treatment of GBM [32]. However, the primary objective of nearly every combination therapy trial is to kill tumor cells. This approach may miss a small population of tumor cells that are dividing inactively and, therefore, not targeted by therapeutics, yet remain capable of active migration. In this way, the use of aptamers—molecules that can be selected to specifically capture migrating cells—presents an intriguing therapeutic platform.

The crypto-aptamer bi-(AID-1-T) appears to be a promising antiproliferative agent, demonstrating considerable versatility across a wide range of high-grade glioma tumor cell cultures characterized by significant heterogeneity [23,33]. This crypto-aptamer has also demonstrated a selective cytostatic effect specifically on GBM cells, in contrast to normal cells, making it highly attractive for further practical investigations [33]. However, it has been shown that while bi-(AID-1-T) decreases proliferation, it does not necessarily inhibit tumor cell migration in certain instances; in fact, it may even enhance migration in some cases [23]. Additionally, the specific protein targeted by this crypto-aptamer remains unidentified. Consequently, this study aims to identify a combination of aptamers capable of simultaneously inhibiting both the proliferation and migration of human GBM cells.

## 2. Results

The observed antiproliferative properties of the crypto-aptamer bi-(AID-1-T) during its cytostatic effect on human glioblastoma tumor cells necessitated the search for its target protein [23,33]. The concept of a crypto-aptamer was introduced by A. Kopylov to define an effective oligonucleotide that does not have a target protein [34]. If such a target is identified, bi-(AID-1-T) can be classified as an aptamer.

### 2.1. Assessment of Bi-(AID-1-T) Specificity to the Target Protein

The search for the target protein bi-(AID-1-T) was performed on human GBM G01 cell culture through the pull-down method, followed by mass spectrometry. Cells were lysed and centrifuged, after which the supernatant was transferred to a protein adsorption column containing avidin-agarose. Following sorption, the column filtrate was divided into three equal parts and transferred into separate tubes: (1) a positive control (biotin-labeled control oligonucleotide), (2) an experimental oligonucleotide (biotin-labeled bi-(AID-1-T)), and (3) a negative control (deionized water) for incubation. After incubation with the oligonucleotides, the solutions were transferred to streptavidin agarose columns. After incubation and washing with high- and low-salt solutions, the samples were eluted from the columns. The resulting samples were analyzed using polyacrylamide gel electrophoresis (PAGE). Comparison of the samples after protein PAGE showed the emergence of two additional bands in sample 2 after the incubation of the cell lysate with biotin-labeled bi-(AID-1-T) (Figure 1).

LCMS was used to analyze bands that are unique to the experimental sample and the corresponding regions in the control sample. The masses of proteins specifically bound to bi-(AID-1-T) that did not interact with the control oligonucleotide were found to range from 33 to 36 kDa and from 110 to 120 kDa, respectively. Mass spectrometric analysis using the software Peaks Studio 10.0 showed that in the 110–120 kDa heavy band, biotin-labeled bi-(AID-1-T) has high and unconditional specificity for the nuclear protein PARP1 (Table 1, Appendix A).

### 2.2. Assessment of PARP1 Affinity to Bi-(AID-1)-T

The biolayer interferometry method was used to verify the specificity of bi-(AID-1-T) (a bi-modular G4 with parallel chain stacking [33]) to PARP1. This method allows testing the affinity of oligonucleotides to the target protein. As negative controls, we used dsDNA oligonucleotide, which was 49 nucleotides long and capable of forming a hairpin structure, and ssDNA oligonucleotide, which was 18 nucleotides long.

Formation of a PARP1 complex with nucleic acids was observed in the case of dsDNA, ssDNA, and bi-(AID-1)-T (Figure 2). However, the binding patterns of PARP1 to the different oligonucleotides differed. For dsDNA and bi-(AID-1)-T, the amplitude of the interference shift signal was significant, about 8 nM, while the signal amplitude for ssDNA was around 1 nM (Figure 2). When the concentration of the tested protein was decreased for both the dsDNA and ssDNA case, a change in the PARP1 concentration to the 20 nM level did not decrease the observed signal amplitude. In contrast, bi-(AID-1)-T exhibited a decrease in signal amplitude when the PARP1 concentration was decreased (Figure 2). The sensitivity of the various oligonucleotides to PARP1 varied significantly, which was a very good result. For ssDNA and dsDNA, the minimum protein concentrations at which the signal was detected were 20 nM, whereas bi-(AID-1)-T produced an association signal already at 5 nM The kinetic rate constant for the association of the bi-(AID-1)-T complex with PARP1 was 6.6 ± 0.34 (10^5^ M^−1^·c^−1^) and the kinetic dissociation rate constant k of 1.28 ± 0.04 (10^2^ c^−1^), giving aKD of 19.4 ± 0.7 nM (Table 2).

Additionally, an experiment was conducted using salmon sperm DNA, a material known for its ability to bind to potential non-specific targets, to achieve more precise probe–target interactions. For this purpose, PARP1 measurement solutions were prepared by adding a standard concentration of 0.1 mg/mL salmon sperm DNA. The interference shift signal completely disappeared in the case of ssDNA in the presence of blocking DNA (Figure 2). In the case of dsDNA, there was a significant reduction in the signal. However, in the case of bi-(AID-1)-T with blocking DNA, the signal amplitude slightly decreased, but interaction was preserved overall.

Based on the measured kinetic parameters of the interaction between bi-(AID-1)-T and PARP1, we computed the aKD of the complex, which is 19.4 ± 0.7 nM and falls within the nanomolar concentration range. Moreover, we first established the preferential order for PARP1 binding. PARP1 binds to single-stranded nucleic acid with low affinity, probably due to charge interactions. PARP1 exhibits a stronger affinity for long double-stranded DNA, since binding to a single-stranded oligonucleotide was completely absent when PARP1 was incubated with blocking DNA. PARP1 interacts with hairpin DNAs more preferentially than with long double-stranded DNA, as evidenced by the presence of a weak signal when analyzing the interaction of DS-DNA with PARP1 in the presence of long blocking DNA. Apparently, the domains that hold PARP1 protein at the DNA damage site are involved in the binding of PARP1 to short hairpin DS-DNA. Binding of the bi-(AID-1)-T oligonucleotide to the PARP1 protein was the most affinity-dependent, with an aKD of 19.4 ± 0.7 nM Moreover, G4 bi-(AID-1)-T retained binding to PARP1 most efficiently in the presence of blocking double-stranded DNA, suggesting that this G4 could be a promising candidate as a high-affinity PARP1 agent.

This is a very exciting finding because the DNA repair protein poly-(ADP-ribose) polymerase 1 (PARP1) is a rather promising target for blocking in cancer. A number of PARP1 inhibitors are presently undergoing clinical trials. For example, multicenter clinical trials with two PARP1 inhibitors rucaparib and olaparib are currently underway. At the same time, similar inhibitors are considered promising in glioblastoma [35]. PARP1 is involved in both DNA repair and maintenance of genome stability [36]. The PARP1 inhibitors being developed bind to the protein’s catalytic domain to block the synthesis of ADP–ribose polymers. As a result, a decrease in the cell’s reparative response to DNA damage is achieved, which triggers the apoptosis cascade [36]. Moreover, it has been demonstrated that PARP1 is upregulated as glioma malignancy increases, and its increased representation in tumor cells allows these cells to repair damaged DNA sequences during their active division [37,38].

### 2.3. G-Quadruplex Bi-(AID-1-C) as an Analog of Bi-(AID-1-T)

In human gliomas, the bi-(AID-1-T) G-quadruplex (GQ) demonstrates potential as an antiproliferative aptamer [33]. However, this aptamer does not capture cells with upregulated Sox2, L1CAM, CD44 [23], while these markers typically associated with stem cells and cells with increased migration levels. This may lead to preservation of the tumor cell population and an increased proportion of cells capable to migrate.

It was found that bi-(AID-1-T) reduced tumor cell migration in some human glioma cell cultures, while in others, it actually increased migration. Therefore, it is evident that additional combined effects on tumor cells should be sought to reduce both the proliferation of glioma cells and their invasive properties.

In the presence of cations, oligonucleotides containing polyguanine blocks of two or more guanines can fold into four-stranded structures—G-quadruplexes. G-quadruplexes are known for their antiproliferative properties in tumors. Bi-(AID-1-T) is a well-known example of a G-quadruplex. It was developed as a dimer of AID-1-T, which is one of the antiproliferative G-quadruplexes composed of simple repetitive sequences (G3T)_4_. Research has revealed that it is more efficient than its monomeric precursor [23,33]. In the GQ DNA T-series, an analog of AID-1-T, consisting of repetitive sequences (G3C)_4_, was also investigated. This G-quadruplex, known as T40214 or STAT [39], has demonstrated its antiproliferative effects on the STAT3 pathway in both prostate cancer [40] and breast cancer [41]. Similar to AID-1-T and bi-(AID-1-T), a dimer of STAT was created, resulting in a new G-quadruplex named bi-(AID-1-C) (Table 3).

The time of penetration of G-quadruplexes bi-(AID-1-T) and bi-(AID-1-C) into the cell and their distribution over time were compared. The analysis was performed on human glioblastoma G01 cell culture.

For this study, two G-quadruplex conjugates, bi-(AID-1-T) and bi-(AID-1-C), each tagged with a fluorescent FAM label at the 3′ end, were synthesized. Time intervals of 1.5, 3, 24, 48, and 72 h, along with a concentration of 1 μM, were used to analyze the penetration of G-quadruplexes into tumor cells. Bi-(AID-1-T) began to accumulate as early as 1.5 h post-addition, reaching its maximum representation by 24 h, and maintaining this accumulation intensity for up to 72 h following exposure. In contrast, the accumulation of the G-quadruplex bi-(AID-1-C) starts only after 3 h, increasing by 24 h, and remained stable for 72 h after co-incubation with tumor cells. Consequently, bi-(AID-1-T) exhibited a gradual accumulation of the aptamer in the cytoplasm and perinuclear space. In comparison, the accumulation intensity of bi-(AID-1-C) was significantly lower, and the distribution of the aptamer within the cytoplasm was uniform, albeit at a slower rate (Figure 3). It could not be excluded that bi-(AID-1-C) is able to exert some extracellular impact, and this requires further thorough study.

G-quadruplexes showed differences in their accumulation and distribution rates within GBM tumor cells, which could indicate different pathways for cellular entry. We conducted an experiment to evaluate whether G-quadruplexes can infiltrate the cell via dynamin-dependent endocytosis. Two previously used human GBM cultures were selected for this study: G-01 and Sus\fP2 [23].

To investigate the role of dynamin-dependent endocytosis in the cellular uptake of bi-(AID-1-T) and bi-(AID-1-C), we utilized Dynasore (Abcam, Cambrige, UK), a small molecule that inhibits the GTPase activity of dynamin-1, dynamin-2, and Drp1. Thus, Dynasore inhibits dynamin-dependent endocytosis by preventing the formation of endocytic vesicles [42,43]. To confirm the efficacy of the inhibitor, we employed staining for the early endosome marker EEA1.

It was observed that in the G01 culture, blocking dynamin significantly reduced the amount of bi-(AID-1-T) aptamer entering the cell (Figure 4A). In contrast, this effect was not observed for Sus\FP2 (Figure 4B). Additionally, no decrease in the cell penetration of bi-(AID-1-C) was noted in any of the cultures (Figure 4). The fluorescence changes of FAM-labeled aptamers are also presented in the Appendix A.

This suggests that different cell penetration pathways may be involved for various glioma cell cultures and different G-quadruplexes. In the G01 culture, bi-(AID-1-T) enters cells via dynamin-dependent endocytosis. Conversely, in the case of bi-(AID-1-C) and the Sus\fP2 culture, dynamin-independent G-quadruplex entry pathways appear to be involved. Further investigation is required to elucidate the mechanisms of bi-(AID-1-T) and bi-(AID-1-C) G-quadruplex penetration into tumor cells.

When examining the role of dynamin-dependent endocytosis in the penetration of G-quadruplexes into cells, it was observed that, in the absence of a blocker, there was some overlap between aptamers and early endosomes. However, since the use of Dynasore necessitates a serum-free medium, the decision was made to analyze the penetration of G-quadruplexes bi-(AID-1-T) and bi-(AID-1-C) into early endosomes and lysosomes in a G01 cell culture under standard conditions, i.e., in the presence of serum in the culture medium. For this purpose, LysoTracker Red (Thermo Fisher Scientific, Waltham, MA, USA) was used, which was added to the cell cultures following incubation with FAM-labeled bi-(AID-1-T) and bi-(AID-1-C) aptamers, after which the cells were fixed. To assess the penetration of the aptamers into early endosomes, the cells were fixed and stained with EEA1 antibodies according to a standard immunocytochemical protocol.

The bi-(AID-1-T) aptamer was found to be slightly present in early endosomes and undetectable in lysosomes, while bi-(AID-1-C) did not penetrate either early endosomes (Figure 5A) or lysosomes (Figure 5B) at 1.5 and 3 h.

Therefore, it can be inferred that bi-(AID-1-T) and bi-(AID-1-C) aptamers can partially enter the cell by dynamin-dependent endocytosis, with bi-(AID-1-T) entering by this route in G01 culture and bi-(AID-1-C) entering by a dynamin-dependent route in Sus\fP2 culture. There was no evidence of bi-(AID-1-T) or bi-(AID-1-C) penetrating early endosomes or lysosomes.

### 2.4. Evaluation of Bi-(AID-1-C) Specificity to the Target Protein

Based on the distinct differences in the mode and rate of penetration of bi-(AID-1-T) and bi-(AID-1-C) into cells, it is expected that their target proteins will also differ. The pull-down method, followed by mass spectrometry, was used to identify the target proteins for bi-(AID-1-C), similar to the approach used for bi-(AID-1-T). The methodology of this study was consistent with prior research. Consequently, biotin-labeled bi-(AID-1-C) was found to interact specifically with cellular proteins in the range 55–75 kDa (Figure 6). The full image is presented in the Appendix A. Subsequent mass spectrometric analysis revealed that bi-(AID-1-C) has high specificity for the nuclear protein RU17 (SNRNP70) (Table 4, Appendix A).

NRNP70 is involved in the chromatin association of hundreds of lncRNAs (long non-coding RNAs) and unstable transcripts in cells, which may play a significant role in tumorigenesis [44]. Since SNRNP70 is a key early regulator of 5′ splice site selection, it is possible that SNRNP70 acts as an effector of alternative splicing of SETMAR mRNA and may serve as a major, albeit not yet fully understood, marker of GBM [45]. There are very few studies examining the relationship between SNRNP70 and tumor processes, so it is quite challenging to assess the role of this protein in malignancy. Nonetheless, we were especially pleased with the data from a study by Dong Jiang et al. This research demonstrated that higher expression levels of SNRNP70 are characteristic of hepatocellular carcinoma (HCC), and that increased expression correlates with a greater likelihood of poor overall survival. Crucially, the authors showed that the knockdown of SNRNP70 inhibited the proliferation and migration of HCC tumor cells [46]. Thus, this aptamer at least merits consideration in the context of its combined effects on migrating GBM tumor cells.

### 2.5. Complex Effects of Bi-(AID-1-T) and Bi-(AID-1-C) Aptamers on Human GBM Tumor Cells

Because of the differences in the mode of entry, distribution, and targeting of bi-(AID-1-T) and bi-(AID-1-C) in the cell, a hypothesis about the possible combined use of these aptamers as antitumor agents has been proposed. The first step to confirm this theory and to determine the most effective addition sequence was a real-time proliferation rate study on the xCELLigence device after exposure of human glioblastoma cell cultures Sus\fP2 and G-01 to various combinations and different exposure sequences of bi-(AID-1-T) and bi-(AID-1-C) was performed. The addition modes are presented in Table 5.

Cell index values were evaluated starting from day 4 after the addition of the first aptamer.

Sequential addition of two aptamers (variations TT, TC, CC, and CT) was found to have the most effective antiproliferative effect on Sus\fP2 culture; triple addition of any combination of the aptamers (TTT, TCT, CCC, CTC) did not yield any advantages. Furthermore, interruptions in the addition of aptamers resulted in diminished effects compared to a single addition (T-T, T-C, C-C, C-T). Simultaneous addition of aptamers (T + C) had a significant antiproliferative effect, but at the same time the cell index began to increase on day 7 of observation (Figure 7A,B).

The antiproliferative effect was less pronounced on the G01 culture; however, similar to Sus\fP2, there was no discernible difference between the twofold and threefold aptamer additions. Both simultaneous and interrupted aptamer additions had a detrimental stimulating effect on cell proliferation. In comparison to a single addition of bi-(AID-1-T), the combinations of TT and TC had less noticeable effects, while the combinations of CC and CT demonstrated more noticeable effects (Figure 7C,D).

Based on the data obtained, four most effective combinations were selected for further study: TT, TC, CC, CT.

The panel of glioma cell cultures (Grade III and IV) was expanded to include five human glioma cultures obtained from the postoperative material of patients. All cultures were classified as Grade III–IV and were identified as IDH1 wild type according to WHO classification (Table 6). 

For the selected combinations, changes in proliferative activity were assessed on a sample of human glioma cultures by MTS test, confirming the data obtained from xCELLigence and proving antiproliferative activity. Although all selected exposure variations exhibited a significant antiproliferative effect, the most pronounced effect was observed with the CC and CT exposure variations (Figure 8A).

However, a significant number of antiproliferative medications are currently under development that enhance the migratory properties of surviving tumor cells. Therefore, we further assessed the effect of selected combinations on the migratory activity of a sample of human high-grade glioma cultures.

The double addition of bi-(AID-1-C) (CC exposure variation) was found to stimulate migratory activity in the majority of cell cultures. In contrast, the double addition of bi-(AID-1-T) (TT exposure variation) and the addition of bi-(AID-1-T) following bi-(AID-1-C) (CT exposure variation) had a significant anti-migratory effect (Figure 8B).

Based on the data obtained, it can be concluded that combined exposure to aptamers is more effective than solo application. The most significant antiproliferative and anti-invasive effect was exerted by double addition of bi-(AID-1-T) and the use of bi-(AID-1-C) followed by addition of bi-(AID-1-T).

All exposure to glioma cells, however, leaves a pool of surviving cells, and future predictions of recurrence heavily depend on the characteristics of these surviving cells. Immunocytochemistry was used to stain cells following a single exposure to bi-(AID-1-T), bi-(AID-1-C), and their most effective combinations of TT and CT. The expression levels of markers associated with stemness and malignancy were evaluated. The G01 culture was chosen for this purpose. On day 4 of the experiment, the expression levels of Nestin and PARP1 decreased with all variations of aptamer addition. Notably, the addition of the TT aptamer led to a decrease in p53 and c-Myc levels, whereas the CT resulted in a decrease in L1CAM, Caveolin-1, and c-Myc levels, which did not occur with the other aptamer variations (Figure 8C–F).

In light of this, the combined use of bi-(AID-1-T) and bi-(AID-1-C) in the variations of double addition of bi-(AID-1-T) (TT variation) and sequential addition of bi-(AID-1-T) and bi-(AID-1-C) (CT variation) not only demonstrated a significant antiproliferative and anti-invasive effect but also resulted in the downregulation of Nestin, PARP1, Caveolin-1, and c-Myc (Figure 8D,E), which can be clearly interpreted as a positive prognostic indicator.

## 3. Discussion

Many known aptamers designed to target specific tumor cell proteins exhibit antiproliferative activity, which is essential for achieving the desired antitumor effects in malignancies. A small panel of aptamers specific to GBM has also been developed. Unfortunately, investigations into the effects of these aptamers are often conducted on established cell lines rather than primary cell cultures, which are more heterogeneous and may respond differently to therapies. This discrepancy contributes to the subsequent failure of clinical trials [47,48,49,50]. Furthermore, for an aptamer to be effective in treating cancer, particularly glioma, it must positively influence the inhibition of cell migration—a factor that is often overlooked in research.

G-quadruplex bi-(AID-1-T) is one of several promising anticancer aptamers [33]. It showed a significant antiproliferative effect on both culture lines of human GBM [33] and on primary cultures of high-grade gliomas derived from postoperative patient material [23]. However, when its effects on migratory activity were examined, it was found to promote cell migration of some primary GBM cell cultures and, unfortunately, does not target tumor stem cells [23].

We became interested in and decided to investigate bi-(AID-1-T) due to its favorable antiproliferative properties on various heterogeneous cell cultures derived from grade III and IV gliomas from patients’ tumor tissue.

During this research, we demonstrated that the PARP1 (poly-(ADP-ribose) polymerase 1) protein turned out to be the binding target for bi-(AID-1-T). PARP1 is a fairly common nuclear protein involved in various cellular processes, including DNA repair, DNA replication, transcription regulation, ribosome biogenesis, programmed cell death, and the maintenance of genomic stability [51,52,53]. It has emerged as a promising target for anti-cancer therapies, leading to extensive research on PARP1 inhibitors [54,55,56]. PARP1 binds to DNA strand breaks and forms a poly-(ADP-ribose) chain of NAD + substrate that turns on cellular cascades to repair DNA damage. The expression of PARP1 mRNA and protein has been shown to be upregulated in several malignancies, including breast, ovarian, skin, colorectal, lung, and brain cancers [57,58,59]. Notably, brain tumors also exhibit a similar upregulation of PARP1 [37,60,61]. For instance, PARP1 is overexpressed in glioblastoma [62], high-grade astrocytomas, medulloblastoma, and pediatric ependymoma [63,64]. It has been shown that PARP1 upregulation enhances tumor resistance to apoptotic processes, contributing to the tumor’s resistance to various therapies, including chemotherapy and radiation therapy [54,55]. An immunohistochemical study of PARP1 representation in human GBM sections revealed that it is upregulated in the nuclei of tumor cells [37,59]. Murnyák et al. showed that 54 out of 60 (90%) GBM tumor samples were positive for PARP1. Furthermore, it was demonstrated that the upregulation of PARP1 correlates with that of p53 and ATRX [37]. The discovery of upregulated PARP1 in glioblastoma tumor stem cells is a significant result of the research. It has been reported that a combination of PARP1 inhibitors and temozolomide can reduce chemoresistance in stem cells. Tentori et al. demonstrated the antitumor effects of this combination on GBM cell lines, showing efficacy in 8 out of the 10 cell lines examined. Additionally, the effectiveness of PARP1 inhibitors as antitumor agents has been established independently of temozolomide [55].

Thus, the specificity of bi-(AID-1-T) to PARP1 makes it a promising candidate for therapy.

The ability of FAM-labeled bi-(AID-1-T) and bi-(AID-1-C) to penetrate tumor cells was investigated. The results indicated that while bi-(AID-1-C)-FAM begins to penetrate cells only after 24 h, bi-(AID-1-T)-FAM does so as early as one hour after addition. Moreover, the accumulation intensity of bi-(AID-1-C)-FAM is significantly lower, suggesting a different pathway for their cellular penetration. Indeed, it has been shown that bi-(AID-1-T) partially penetrates the cell via the dynamin-dependent endocytosis pathway, whereas the penetration of bi-(AID-1-C) into the cell is independent of dynamin activity and occurs through alternative mechanisms that require further study.

Given the marked differences in cell penetration rates and mechanisms, the target protein for bi-(AID-1-C) should also differ. In mass spectrometric analyses, the bi-(AID-1-C) G-quadruplex demonstrated high specificity to the nuclear protein SNRNP70. This protein is not yet widely popular as a target of interest in GBM therapy. However, SNRNP70 is highly expressed in hepatocellular carcinoma (HCC), and increased expression correlates with poorer patient outcomes. Moreover, SNRNP70 knockdown has been shown to inhibit HCC cell proliferation and migration [46]. SNRNP70 possesses a helix–loop–helix domain and possibly causes constitutive activation of NTRK3 through dimerization, which may play a role in tumorigenesis [65]. Lié et al., who studied 47 GBM samples, are finally beginning to link SNRNP70 to the tumor of our interest [45]. It is known that snRNP70 binds to U1 spliceosome RNA to form U1snRNP, a major component of the spliceosome [66], and U1snRNP is associated with tumor cell migration [67].

PARP1 and SNRNP70 play crucial roles in cellular processes, including RNA processing and DNA damage response. Both proteins are highly expressed in malignant diseases, which makes them attractive targets for antitumor intervention. Consequently, our preliminary findings led us to the decision to use bi-(AID-1-T) and bi-(AID-1-C) in combination. 

The concept of combined effects on tumors is gaining traction, as tumors have been shown to be heterogeneous. This heterogeneity explains why certain tumor cells survive after monotherapy, particularly in the case of heterogeneous GBMs. Therefore, researchers are exploring combinations of therapeutic molecules capable of targeting as many tumor cells as possible. For example, Kucinska et al. demonstrated that combinations of dacomitinib and foretinib, dacomitinib and DSF/Cu, and foretinib and AZD3759 are more effective in the treatment of GBM [68]. However, as previously mentioned, all of these agents have cytotoxic effects, which pose a risk to healthy cells as well. An aptamer with its cytostatic action can be a less traumatic treatment option. Therefore, it is essential to identify an aptamer combination that not only inhibits the proliferation of GBM cells but also affects their migration. 

There are already documented cases of several aptamers; however, they are primarily utilized for diagnostic purposes to enhance visualization. For instance, Kim et al. and Cao et al. demonstrated that a mixture of aptamers can be used to increase the sensitivity of bacterial cell detection [69,70]. Chang et al. and Cai et al. proposed using combinations of aptamers for imaging circulating tumor cells. For therapeutic purposes, aptamers are typically employed for targeted delivery by combining them with existing drugs [71].

In turn, we suggest that the two aptamers, bi-(AID-1-T) and bi-(AID-1-C), be combined as antitumor agents to inhibit both the proliferation and migration of human GBM tumor cells. 

On human glioblastoma G01 and Sus\fP2 cell cultures, the greatest time-consistent antiproliferative effect was observed with the sequential addition of two aptamers, administered without interruption between the additions (Figure 7). These addition variations were TT (bi-(AID-1-T) and bi-(AID-1-T)), TC (bi-(AID-1-T) and bi-(AID-1-C)), CC (bi-(AID-1-C) and bi-(AID-1-C)), and CT (bi-(AID-1-C) and bi-(AID-1-T)). These selected combinations were tested on an expanded sample of human GBM cultures derived from postoperative patient material. All combinations demonstrated a strong antiproliferative effect; however, only two of them—TT and CT—were found to significantly reduce migratory activity across all tested cultures (Figure 8).

The most effective combinations of TT and CT significantly reduced both the proliferation and migration of human GBM tumor cells, although a small pool of surviving cells remained. However, these cells do not possess stemness or malignancy markers unlike with other therapeutic options. Both combinations demonstrated a significant downregulation of Nestin, PARP1, and c-Myc. Additionally, the TT variation resulted in downregulation of p53, while the CT variation led to downregulation of L1CAM and Caveolin-1 (Figure 8C). These changes in tumor cell characteristics suggest that the TT and CT combinations may target a subset of tumor stem cells and reduce tumor malignancy.

Nestin is widely recognized as a neuroepithelial stem cell marker and is frequently upregulated in gliomas, with its expression increasing in correlation with the degree of malignancy [72]. It plays an important role in tumor cell growth, migration, and invasion processes [73]. Some studies have indicated that increased Nestin expression is associated with a poorer prognosis in glioma patients [74,75]. The knockdown of Nestin in human GBM cells has been shown to suppress cell proliferation, migration, and invasion [76]. Reduced levels of Nestin can disrupt the spindle apparatus during GBM cell division, ultimately leading to cell death [77].

PARP1, whose role in cancer we emphasized earlier, is also significantly reduced when exposed to combinations of aptamers. This is undoubtedly favorable and suggests a positive prognosis.

It is believed that the oncogene c-Myc regulates transcription, which subsequently controls stem cell growth [78,79]. c-Myc is found in many human tumors, correlates with the degree of malignancy and is often associated with a poor prognosis [80,81,82]. c-Myc inhibitors are being widely developed to improve antitumor therapy [83,84]. The inhibition of c-Myc reduces the proliferation and increases the apoptosis of glioma cells [82].

All of these markers are overexpressed in glioblastomas, and their inhibition and downregulation reduce the proliferation and migration of tumor cells. This is supported by our findings and may also increase the tumor’s sensitivity to additional therapies.

In addition, it is important to note that the TT combination leads to the downregulation of p53. The well-known tumor suppressor p53 has been associated with improved patient outcomes when its expression levels are low [85,86,87]. The survival of p53-positive cells was observed when bi-(AID-1-T) was applied alone to a sample of cultures [23]; however, it was significantly downregulated when the TT combination was used.

However, it is particularly noteworthy that the CT combination additionally reduces the expression levels of L1CAM and Caveolin-1.

Cell adhesion molecule L1CAM is expressed in high-grade gliomas and is widely associated with tumor cell migration, invasion, and resistance to apoptosis [88,89,90]. Additionally, L1CAM is considered a marker for tumor stem cells and is believed to enhance their resistance [90,91].

The role of Caveolin-1 in tumors is quite ambiguous; it can act both as a tumor suppressor and, conversely, promote tumor growth, depending on the tumor type and its stage of progression [92,93,94]. However, Caveolin-1 is highly represented in gliomas and has an unambiguous negative correlation with patient survival [95,96]. It can interact with various signaling molecules, thereby influencing glioma development and progression [97]. Additionally, Caveolin-1 positively correlates with markers of tumor migration, enhances tumor cell resistance to therapy, and maintains stemness [98].

Thus, the CT demonstrated the most favorable results, showing a significant reduction proliferation and migration of GBM cells. Additionally, it resulted in a consistent decrease in stemness and migration markers, which was not observed in the other samples.

These findings demonstrate that using several aptamers in succession can target multiple mechanisms and induce a persistent antitumor effect on human GBM cells.

## 4. Material and Methods

### 4.1. Cell Cultures

Primary cultures were obtained from human postoperative glioma complying with all formal requirements of the Russian Federation. This study was approved by the Ethics Committee of Burdenko Neurosurgical Institute, Russian Academy of Medical Sciences (№_12/2020). All subjects gave written informed consent in accordance with the Declaration of Helsinki. The cells were cultured in DMEM/F12 medium (Servicebio, Wuhan, Hubei, China) supplied with 10% FBS (Biowest, Nuaille, France), 2 mM L-glutamic acid, (Paneco, Moscow, Russia), and 1% antibiotic solution (penicillin/streptomycin) (Corning, Corning, NY, USA) at 37 °C and 5% CO_2_. Cells were removed from culture vessels using Versene solution (Paneco, Russia) and 0.25% Trypsin solution (Paneco, Russia).

### 4.2. Assessment of Bi-(AID-1-T) Specificity to the Target Protein

#### 4.2.1. Lysate Preparation and Pull-Down

A G01 glioblastoma cell culture was cultivated until 80% confluency was achieved, then trypsinized with 0.25% trypsin solution (Paneco, Russia) and washed with PBS. IP buffer (25 mM Tris-HCl pH 7.4, 150 mM NaCl, 1 mM EDTA, 1% NP-40, 5% glycerol, 0.1 mM DTT, and protease inhibitor) was added to the cell sediment on ice, at a rate of 500 µL buffer per 10 million cells. The lysate was centrifuged for 30 min at 21,000× *g* at 4 °C. The supernatant was transferred to a prepared protein sorption column (150 µL Pierce Avidin Agarose (ThermoFisher Scientific, USA) applied to Pierce Spin Columns (ThermoFisher Scientific, USA), then washed three times with PBS), added RNase A (ThermoFisher Scientific, USA) and incubated at room temperature on a shaker for one hour. After sorption, the column filtrate was divided into equal parts and transferred into tubes containing (1) positive control, biotin-labeled control oligonucleotide (biotin-GGACGGGGATTTAATCGCCGTAGTAGAAAAGCATGTCAAAAGCCGGAACCGTCC) (2.5 µL of 100 µM solution) (GenTerra Ltd., Moscow, Russia), (2) experimental oligonucleotide-biotin-labeled bi-(AID-1-T) (biotin-GGGTGGTGGGTGGTGGTGGTGGTGGTGGTGGTGGTGGTGGTGGTGGTGGTGGTGGTGGG) or bi-(AID-1-C) (biotin-GGGCGGCGGCGGCGGCGGCGGCGGCGGCGGCGGCGGCGGG) (2.5 µL 100 µM solution) (GenTerra Ltd., Russia), (3) negative control—deionized water. After incubation on a shaker for 2 h at room temperature, the solutions were transferred to the prepared columns (40 μL of 50% suspension of Pierce Streptavidin Agarose (ThermoFisher Scientific, USA) was applied to the columns, washed three times with PBS) and incubated on a shaker for 40 min at room temperature. Next, 500 μL of IP buffer was applied to the columns, centrifuged, and washed three times sequentially with high-salt buffer (25 mM Hepes, 1mM EDTA, 0.1% NP40, 300 mM NaCl) and low-salt buffer (25 mM Hepes, 1mM EDTA, 0, 1% NP40, 50 mM NaCl). Proteins were then eluted from the columns two times sequentially with one-time Laemmli buffer (containing 1% SDS) with heating to 85 °C. Samples after the first and second elutions were analyzed using polyacrylamide gel electrophoresis (according to Laemmli) and then stained with silver. Bands unique to the experimental sample and corresponding regions in the control sample were excised, de-stained, and analyzed by LCMS. Illustration of this protocol is provided in the Appendix A.

#### 4.2.2. Chromatography–Mass Spectrometry Analysis

Protein hydrolysis in a polyacrylamide gel using trypsin was performed according to the protocol described previously by Shevchenko et al. [99]. Samples were loaded onto a laboratory-made 20 × 0.1 mm pre-column packed with Inertsil ODS3 3 µm sorbent (GL Sciences, Torrance, CA, USA) in a solution containing 2% acetonitrile, 98% H_2_O, 0.1% THFU, at a flow rate of 10 μL/min and separated at room temperature on a 300 × 0.1 mm fused silica column with an emitter made on a P2000 Laser Puller (Sutter, Novato, CA, USA) and packed in the laboratory Reprosil PUR C18AQ 1.9 sorbent (Dr. Maisch, Ammerbuch, Germany). Reversed-phase chromatography was performed on an Ultimate 3000 Nano LC System chromatograph (Thermo Fisher Scientific) coupled to an Orbitrap Lumos Tribrid mass spectrometer (Thermo Fisher Scientific) via a nanoelectrospray source (Thermo Fisher Scientific). Solvent systems A (99.9% water, 0.1% formic acid) and B (19.9% water, 0.1% formic acid, 80% acetonitrile) were used for chromatographic separation of peptides. Peptides were eluted from the column using a linear gradient: 3–35% B for 58 min, 35–99% B for 5 min, 99% B for 5 min, and 99–3% B for 0.1 min at a flow rate of 500 nL/min. The following instrument settings were used in the mass spectrometric analysis: MS1 scan: resolution 60,000, scan range: 500–1600 m/z, maximum ion injection time—30 ms, AGC level—3 × 10^6^, ion fragmentation was performed in HCD mode at collision energy 30%, MS2 ion registration was performed at resolution 15,000. *Analysis of mass spectrometry data.* Mass spectrometry data were analyzed using Peaks studio 10.0 computer program (Bioinformatics Solutions Inc., Waterloo, Canada). Protein identification was performed by correlating tandem mass spectra with the Uniprot-SwissProt human protein sequence database (23 February 2019) with the following parameters: constant modification of Cys—carbamidomethylation, variable modifications—oxidation of Met and acetylation of the N-terminal amino acid residue of the protein, acceptable level of false positive peptide identifications—0.01 (determined from the reverse amino acid sequence database), protease specificity—C-terminal Arg and Lys (up to two missing hydrolysis sites were allowed when searching the database). In peptide identification, deviation of the experimentally obtained peptide mass from its theoretical mass was allowed up to 10 m.d., and deviation of fragment masses was allowed up to 0.02 Da.

### 4.3. Assessment of PARP1 Affinity to Bi-(AID-1-T)

Oligonucleotides labeled with biotin at the 5′-end were used: bi-(AID-1)-T (biotin-GGGTGGGTGGGTGGGTTTGGGTGGGTGGGTGGG), the control double-strand DNA duplex (biotin-GCCGGCATTTTGACGCCGCCCCGGCTGCTTATGCTCCGGGGCATATGGC), control single-strand DNA oligonucleotide (biotin-CATTTAGGACCAACACAA). Oligonucleotides were purchased from GenTerra Ltd. (Moscow, Russia).

Oligonucleotides were performed at 2 μM reaction mix consisting of Tris-HCl buffer pH 7.4 supplemented with 50 mM NaCl, 10 mM KCl, and 5 mM MgCl_2_: the solution was heated for 5 min to 95 °C followed by slow cooling to room temperature.

*Biolayer interferometry*. A Sartorius Octet R2 instrument (Sartorius, Göttingen, Germany) was used to conduct multi-concentration binding affinity experiments at 25 °C, using 1 × PBS (pH 7.4). SAX Biosensors (ForteBio, 18-5117) were employed for immobilizing biotin-labeled oligonucleotides at a concentration of 2 μM; they were allowed to load on biosensors for 10 min. The recombinant human PARP1 was kindly provided by Prof. Lavrik O.I. (Institute of Chemical Biology and Fundamental Medicine SB RAS). BSA was purchased from Helicon Ltd. (Moscow, Russia), salmon sperm DNA (Sigma-Aldrich, Burlington, MA, USA) was sonicated, and the stock solution had a concentration of 1 mg/mL. Protein was dissolved in cold PBS; the concentration of the stock solution was 0.1 mg/mL. For association phase monitoring, protein samples were prepared by serial dilution of stock solution in phosphate buffer to concentrations of 162 nM, 81 nM, 41 nM, 20 nM, 10 nM, 5 nM, and 2.5 nM The concurrent association experiment included the samples with 0.1 mg/mL salmon sperm DNA and with the same protein concentration. Sensor regeneration was with 1 M ethanolamine hydrochloride solution pH 8.5, while phosphate buffer solution was used at the washing stage. In the Octet data analysis software, data were retrieved via a parallel buffer blank subtraction. The Langmuir model, which specifies a 1:1 binding stoichiometry, was used to fit the computed binding curves using the Origin 2021 program (OriginLab, Northampton, Massachusetts, USA).

### 4.4. Cellular Uptake and Localization of Aptamers

G01 и Sus\fP2 cell cultures were plated on Coverslips 20009 (SPL Life Sciences, Gyeonggi-do, Korea) on 24-well plates (SPL Life Sciences, Korea) at a seeding density of 20,000 cells/well in media and allowed to grow for 24 h. Post attachment, live cells were treated with 1 µM of FAM-bi-(AID-1-T) and FAM-bi-(AID-1-C) (GenTerra Ltd., Russia). After 1.5, 3, 24, 48, and 72 h of incubation, cells were fixed with 4% paraformaldehyde, stained with Hoechst 33342 (Sigma-Aldrich, Burlington, MA, USA) dissolved in PBS, and analyzed using Carl Zeiss confocal microscope of LSM-710 series (Oberkochen, Germany).

### 4.5. Endocytosis in Cellular Uptake of Aptamers

G01 and Sus\fP2 cells were plated on coverslips as it was described above. Standard culture medium was replaced with serum-free medium with 100 µM Dynasore (ab120192, Abcam, Cambrige, UK) and cells were incubated for 30 min, followed by the addition of FAM-bi-(AID-1-T) and FAM-bi-(AID-1-C) (GenTerra, Russia) at a concentration of 1 µM to the medium. After 1.5 or 3 h of incubation, cells were fixed with 4% paraformaldehyde, stained with EEA1 antibodies (ab109110, Abcam, USA) and with Hoechst 33342 (Sigma, USA), and analyzed using a Carl Zeiss confocal microscope of LSM-710 series. Cells in serum-free medium without Dynasore treatment but supplemented with FAM-bi-(AID-1-T) or FAM-bi-(AID-1-C) were used as controls. The experiment was conducted in three repeats.

### 4.6. Aptamer Localization in Early Endosomes and Lysosomes

G01 and Sus\fP2 cells were plated on coverslips as described above. To the standard culture medium with serum, 1 μM FAM-bi-(AID-1-T) or FAM-bi-(AID-1-C) were added and cells were incubated for 1.5–3 h. After that, a part of the cells was treated with 80 nM LysoTracker Red DND-99 (ThermoFisher Scientific, USA) for 10 min, followed by fixation with 4% paraformaldehyde. The other part of the cells was fixed and stained with EEA1 antibodies (ab109110, Abcam, USA). All cells were additionally stained with Hoechst 33342 (Sigma, USA) and analyzed using a Carl Zeiss confocal microscope of the LSM-710 series.

### 4.7. Treatment with Aptamers

Before use, the aptamers were performed by incubating 100 µM aptamer solution at 95 °C for 5 min in a buffer (10 mM KCl, 140 mM NaCl, 20 mM Tris-HCl) with subsequent cooling at room temperature. Afterward, it was diluted with culture medium to a concentration of 10 μM and added to cell cultures according to Table 4.

### 4.8. Cell Viability Assessment by xCELLigence RTCA

G01 and Sus\fP2 cells were plated on E-plates 16 for xCELLigence RTCA (Agilent, Santa Clara, CA, USA) at a density of 7000 cells per well. After cell attachment, aptamers were added to the wells according to Table 4. Using RTCA Software 2.1, the cell index was determined for each variant of aptamer addition on days 4, 5, 6, and 7 of observation. Wells without the addition of aptamers were used as controls.

### 4.9. Cell Proliferation Assessment by MTS Assay

Human high-grade glioma cell cultures (Table 5) were planted on 96-well plates (Nest, China) at 7000 cells/well. After cell attachment, combinations of aptamers that appeared to be most effective in cell viability assessment using xCELLigence RTCA (Agilent, Santa Clara, CA, USA) were added. Then, 72 h after the first aptamer was added, 1/10 of the medium volume of MTS reagent (Promega, USA) was added to the wells. After 90 min incubation at 37 °C and 5% CO_2_, the absorbance was measured by SPECTROstar Nano (BMG LABTECH, Ortenberg, Germany) at 495 nm.

### 4.10. Cell Migration Assessment by Transwell Inserts

Cell migration activity after aptamer exposure was assessed using Transwell inserts with 8 μm pores (353097, Corning Falcon, Corning, NY, USA). Cells were incubated in a serum-free medium for one hour, and then 20,000 cells per well in 350 µL of serum-free medium were transferred into each insert. The corresponding wells contained 700 μL of culture medium with 10% FBS. After 20 h, non-migrated cells on the inner side of the insert were removed, and migrated cells on the outer side of the insert were fixed with 4% paraformaldehyde and stained with Hoechst 33342 (Sigma, USA). The migration activity was defined by cell count in six non-crossing visual fields. Wells without the addition of aptamers were used as controls.

### 4.11. Immunocytochemistry (ICC)

Cells were seeded on Coverslips 20009 (SPL Life Sciences, Korea) on 24-well plates (SPL Life Sciences, Republic of Korea) at a density of 40,000 cells per well and incubated at 37 °C and 5% CO_2_ for 24 h. After exposure to aptamer combinations, cells were fixed with 4% paraformaldehyde and sequentially stained with primary antibodies to CD44 (ab157107, Abcam, USA), Nestin (MA1-110, Invitrogen, Thermo Fisher Scientific, Carlsbad, CA, USA), PARP1 (ab794586, Abcam, USA), L1CAM (ab24345, Abcam, USA), Sox2 (ab97959, Abcam, USA), p53(ab1101, Abcam, USA), Caveolin-1 (MA3-600, Invitrogen, USA), c-Myc(FNab01791, FineTest, China) and secondary antibodies labeled Cy2 (715-225-151, Jackson ImmunoResearch, UK), or Alexa Fluor^®^ 594 (711-585-152, Jackson ImmunoResearch, West Grove, PA, UK). Cell nuclei were stained with Hoechst 33342 (Sigma, USA). Cells were fixed in Mowiol 4-88 (9002-89-5, Sigma-Aldrich, USA) with 1% DABCO for 8 h at 4 °C, then for 24 h at room temperature. The samples were analyzed using a Carl Zeiss confocal microscope of the LSM-710 series and Nexcope NIB 900 fluorescent microscope (Nexcope, Novel Optics, Ningbo, Zhejiang, China). Cell cultures stained with secondary antibodies without primary antibodies were used as antibody controls. The average fluorescence intensity of cells was determined using the ImageJ v1.53 software.

### 4.12. Statistics

Statistical analysis (ANOVA test) was reformed with GraphPad Prism 9.0.0 software. The data are presented as means ± SD from a minimum of three repeats. Statistical significance is presented as * *p* < 0.05, ** *p* < 0.01, *** *p* < 0.001 and **** *p* < 0.0001

## Figures and Tables

**Figure 1 pharmaceuticals-17-01435-f001:**
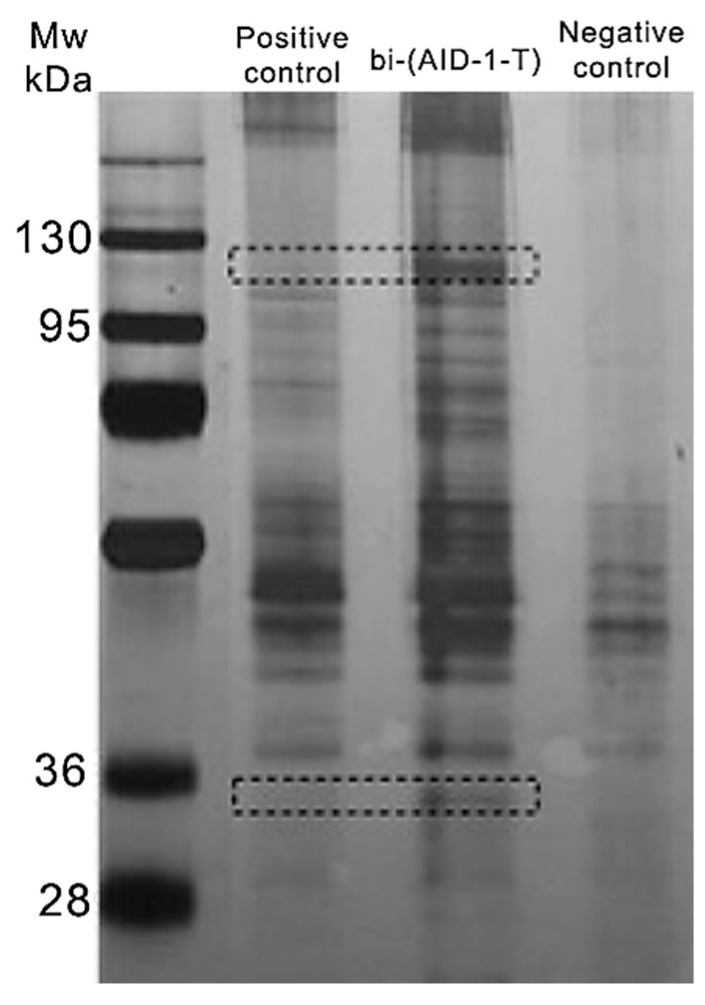
5/10% PAGE of proteins obtained after incubation of cell lysates with biotin-labeled control oligonucleotide and biotin-labeled bi-(AID-1-T) after column elution. The negative control contains oligonucleotide-untreated protein lysate after a second elution. The dotted line indicates excised areas of the gel that were further analyzed.

**Figure 2 pharmaceuticals-17-01435-f002:**
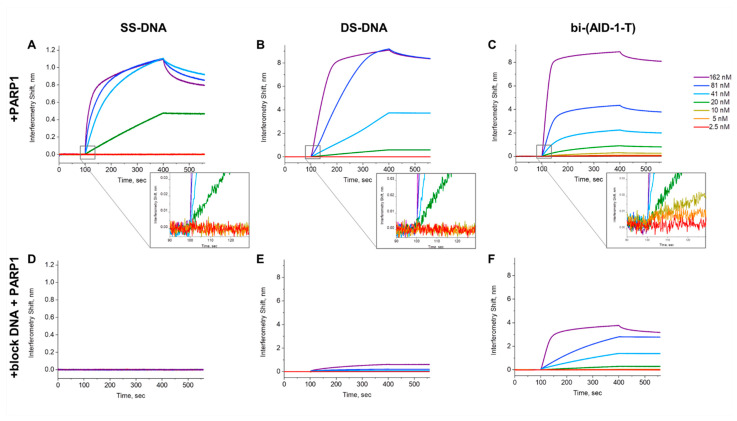
Data on interferometry biolayer interaction of recombinant PARP1 protein at 162, 81, 41, 20, 10, 5 and 2.5 nM concentrations (**A**–**C**), or PARP1 at the same concentrations in the presence of blocking double-stranded DNA at 0.1 mg/mL (**D**–**F**) with different types of oligonucleotides: SS-DNA single-stranded oligonucleotide (**A**,**D**), structured double-stranded oligonucleotide DS-DNA (**B**,**E**), and bi-(AID-1-T) (**C**,**F**). The association initiation region is shown enlarged for figures (**A**–**C**).

**Figure 3 pharmaceuticals-17-01435-f003:**
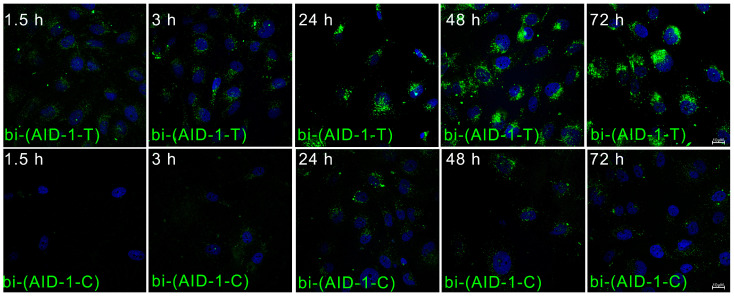
Accumulation and distribution of FAM-bi-(AID-1-T) (top row) and FAM-labeled bi-(AID-1-C) (bottom row) G-quadruplexes in human GBM G-01 culture cells at 1.5, 3, 24, 48, and 72 h after their addition at a concentration of 1 μM.

**Figure 4 pharmaceuticals-17-01435-f004:**
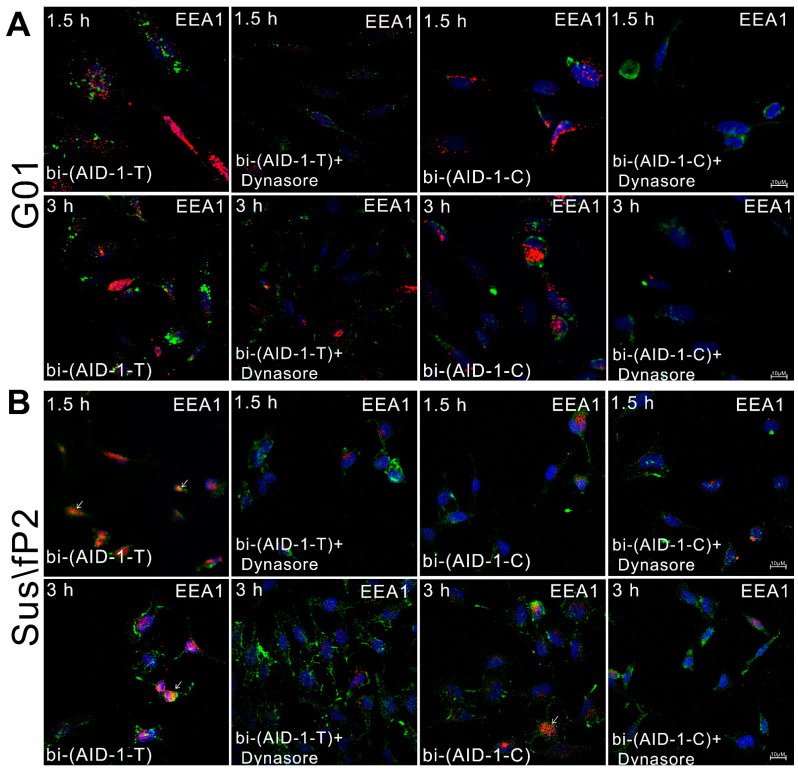
Effect of endocytosis blocker Dynasore on the distribution of FAM-labeled bi-(AID-1-T) and bi-(AID-1-C) aptamers (green) and assessment of aptamer penetration into early endosomes (EEA1-red) of human glioma culture cells G-01 (**A**) and Sus\fP2 (**B**), nuclei stained with Hoechst 33342 (blue). Arrows indicate the overlap of FAM-labeled aptamers and early endosome staining sites.

**Figure 5 pharmaceuticals-17-01435-f005:**
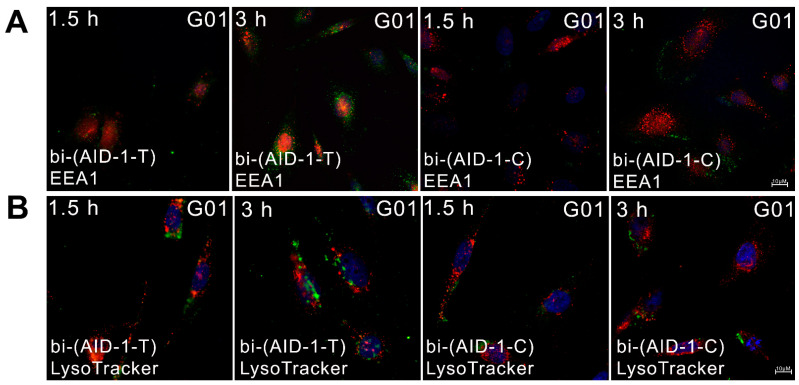
Relative location of FAM-labeled aptamers bi-(AID-1-T) and bi-(AID-1-C) (green) relative to early endosomes (EEA1-red) (**A**) and lysosomes (LysoTracker-red) (**B**) in human glioma culture G-01 cells, nuclei stained with Hoechst 33342 (blue).

**Figure 6 pharmaceuticals-17-01435-f006:**
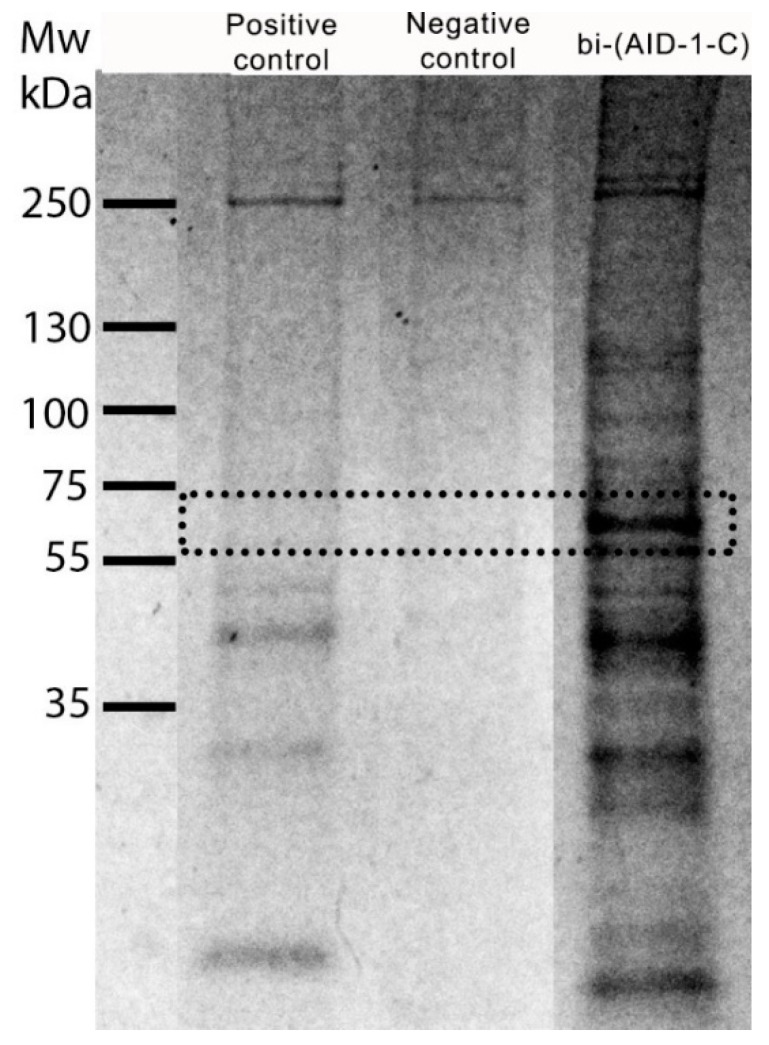
4–20% PAGE electrophoresis of proteins obtained after incubation of cell lysates with biotin-labeled control oligonucleotide (positive control) and biotin-labeled bi-(AID-1-C) after the first column elution. The negative control contains oligonucleotide-untreated protein lysate after the first elution. The dotted line shows excised regions of the gel that were further analyzed.

**Figure 7 pharmaceuticals-17-01435-f007:**
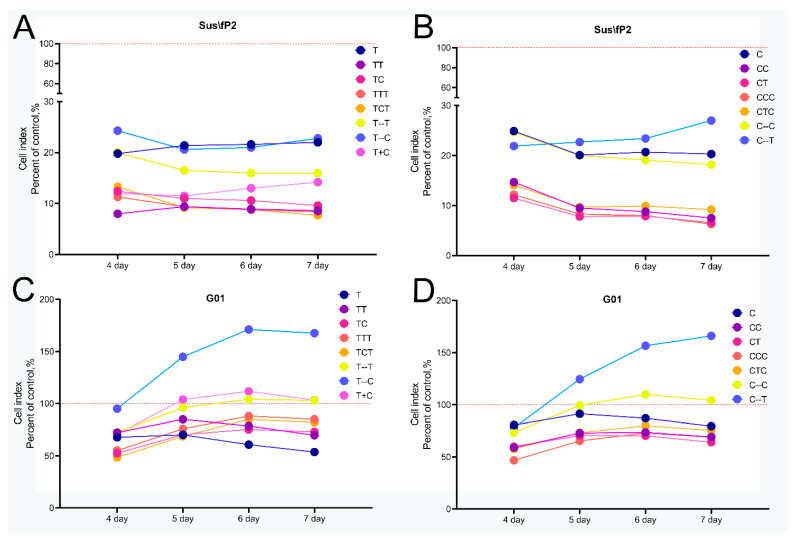
Effect of G-quadruplexes bi-(AID-1-T) and bi-(AID-1-C) and their combinations on changes in the cellular index of human glioma cell cultures Sus\fP2 (**A**,**B**) and G-01 (**C**,**D**) 72 h after addition of the first of the aptamers to the cell cultures; aptamer concentration 10 μM; data are presented as a percentage of the control (as a control, cell cultures without the addition of aptamers were used).

**Figure 8 pharmaceuticals-17-01435-f008:**
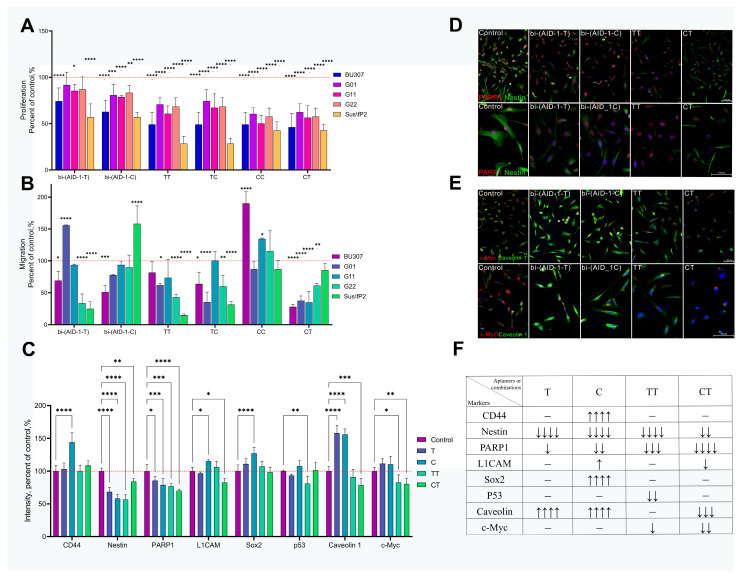
Effect of G-quadruplexes bi-(AID-1-T) and bi-(AID-1-C) and their combinations on human glioma cell cultures BU307, G-01, G-11, G-22, Sus\fP2. (**A**)—Evaluation of proliferative activity by MTS test 72 h after addition of the first of the aptamers to the cell cultures; aptamer concentration 10 μM; data are presented as a percentage of the control (untreated cell culture). (**B**)—Migration evaluation using transwell chambers 72 h after addition of the first of the aptamers to the cell cultures; aptamer concentration 10 μM; data are presented as a percentage of the control (untreated cell culture). (**C**)—Immunocytochemistry (ICC) staining of the G01 cell culture after addition of G-quadruplexes bi-(AID-1-T) and bi-(AID-1-C) and their combinations TT and CT 72 h after addition of the first of the aptamers to the cell culture; aptamer concentration 10 μM; data are presented as a percentage of control (untreated cell culture). (**D**)—ICC staining of G01 cell culture for Nestin (green) and PARP1 (red) after addition of G-quadruplexes bi-(AID-1-T) and bi-(AID-1-C) and their combinations TT and CT 72 h after addition of the first of the aptamers to the cell culture; aptamer concentration 10 μM. nuclei stained with Hoechst 33342 (blue), (**E**)—ICC staining of the G01 cell culture for Caveolin-1 (green) and c-Myc (red) after addition of G-quadruplexes bi-(AID-1-T) and bi-(AID-1-C) and their combinations of TT and CT 72 h after addition of the first of the aptamers to the cell cultures; aptamer concentration 10 μM. nuclei stained with Hoechst 33342 (blue), (**F**)—Degree of change in the expression level of stemness and malignancy markers in G01 cell culture after addition of G-quadruplexes bi-(AID-1-T) and bi-(AID-1-C) and their combinations of TT and CT; * *p* < 0.05, ** *p* < 0.01, *** *p* < 0.001 and **** *p* < 0.0001.

**Table 1 pharmaceuticals-17-01435-t001:** Top 10 proteins specifically bound to biotin-labeled bi-(AID-1-T).

Protein ID	Accession	−10lgP	Coverage (%)	Area Sample	Peptides	Unique	Spec Sample	Avg. Mass	Description
P09874	PARP1_HUMAN	219.24	36	1.60 × 10^8^	60	60	114	113,084	Poly [ADP-ribose] polymerase 1 OS = Homo sapiens OX = 9606 GN = PARP1 PE = 1 SV = 4
Q9H2U1	DHX36_HUMAN	211.11	28	1.44 × 10^8^	47	47	101	114,760	ATP-dependent DNA/RNA helicase DHX36 OS = Homo sapiens OX = 9606 GN = DHX36 PE = 1 SV = 2
Q1KMD3	HNRL2_HUMAN	166.63	27	7.70 × 10^7^	29	29	92	85,105	Heterogeneous nuclear ribonucleoprotein U-like protein 2 OS = Homo sapiens OX = 9606 GN = HNRNPUL2 PE = 1 SV = 1
P68371	TBB4B_HUMAN	125.29	16	0	7	1	24	49,831	Tubulin beta-4B chain OS = Homo sapiens OX = 9606 GN = TUBB4B PE = 1 SV = 1
Q07157	ZO1_HUMAN	119.68	7	2.47 × 10^6^	15	13	18	195,457	Tight junction protein ZO-1 OS = Homo sapiens OX = 9606 GN = TJP1 PE = 1 SV = 3
P16615	AT2A2_HUMAN	114.27	12	2.83 × 10^6^	14	14	16	114,757	Sarcoplasmic/endoplasmic reticulum calcium ATPase 2 OS = Homo sapiens OX = 9606 GN = ATP2A2 PE = 1 SV = 1
Q9UDY2	ZO2_HUMAN	112.37	14	2.50 × 10^6^	21	19	24	133,958	Tight junction protein ZO-2 OS = Homo sapiens OX = 9606 GN = TJP2 PE = 1 SV = 2
P08621	RU17_HUMAN	108.03	19	1.86 × 10^6^	9	9	13	51,557	U1 small nuclear ribonucleoprotein 70 kDa OS = Homo sapiens OX = 9606 GN = SNRNP70 PE = 1 SV = 2
Q9NR30	DDX21_HUMAN	104.2	18	2.01 × 10^6^	15	12	17	87,344	Nucleolar RNA helicase 2 OS = Homo sapiens OX = 9606 GN = DDX21 PE = 1 SV = 5
Q12797	ASPH_HUMAN	103.01	17	3.38 × 10^6^	14	14	18	85,863	Aspartyl/asparaginyl beta-hydroxylase OS = Homo sapiens OX = 9606 GN = ASPH PE = 1 SV = 3

**Table 2 pharmaceuticals-17-01435-t002:** Affinity parameters for bi-(AID-1-T).

Affinity Parameter	Kinetic Association Constant, k_on_ (×10^5^ M^−1^⋅s^−1^)	Kinetic Dissociation Constant, k_off_ (×10^2^ s^−1^)	Apparent Dissociation Constant, aK_D_ nM
bi-(AID-1-T)	6.6 ± 0.34	1.28 ± 0.04	19.4 ± 0.7

**Table 3 pharmaceuticals-17-01435-t003:** Nucleotide sequence of G-quadruplexes bi-(AID-1-T) and bi-(AID-1-C).

Name	Sequence
bi-(AID-1-T)	GGG TGG GTG GGT GGG TTT GGG TGG GTG GGT GGG
bi-(AID-1-C)	GGG CGG GCG GGC GGG TTT GGG CGG GCG GGC GGG

**Table 4 pharmaceuticals-17-01435-t004:** Top 10 proteins specifically bound to biotin-labeled bi-(AID-1-C).

Protein ID	Accession	−10lgP	Coverage (%)	Area Sample	Peptides	Unique	Spec Sample	Avg. Mass	Description
P08621-2	RU17_HUMAN	146.32	25	6.95 × 10^8^	9	9	27	50,618	Isoform 2 of U1 small nuclear ribonucleoprotein 70 kDa OS = Homo sapiens OX = 9606 GN = SNRNP70
P08621	RU17_HUMAN	146.32	24	6.95 × 10^8^	9	9	27	51,557	U1 small nuclear ribonucleoprotein 70 kDa OS = Homo sapiens OX = 9606 GN = SNRNP70 PE = 1 SV = 2
O76021	RL1D1_HUMAN	92.3	11	6.86 × 10^7^	5	5	6	54,973	Ribosomal L1 domain-containing protein 1 OS = Homo sapiens OX = 9606 GN = RSL1D1 PE = 1 SV = 3
O60506-2	HNRPQ_HUMAN	66.87	4	1.15 × 10^7^	3	3	3	65,682	Isoform 2 of Heterogeneous nuclear ribonucleoprotein Q OS = Homo sapiens OX = 9606 GN = SYNCRIP
O60506	HNRPQ_HUMAN	66.87	4	1.15 × 10^7^	3	3	3	69,603	Heterogeneous nuclear ribonucleoprotein Q OS = Homo sapiens OX = 9606 GN = SYNCRIP PE = 1 SV = 2
O43390-4	HNRPR_HUMAN	66.87	4	1.15 × 10^7^	3	3	3	59,953	Isoform 4 of Heterogeneous nuclear ribonucleoprotein R OS = Homo sapiens OX = 9606 GN = HNRNPR
O43390	HNRPR_HUMAN	66.87	4	1.15 × 10^7^	3	3	3	70,943	Heterogeneous nuclear ribonucleoprotein R OS = Homo sapiens OX = 9606 GN = HNRNPR PE = 1 SV = 1
O43390-2	HNRPR_HUMAN	66.87	4	1.15 × 10^7^	3	3	3	71,214	Isoform 2 of Heterogeneous nuclear ribonucleoprotein R OS = Homo sapiens OX = 9606 GN = HNRNPR
P27694	RFA1_HUMAN	62.93	6	5.18 × 10^6^	3	3	5	68,138	Replication protein A 70 kDa DNA-binding subunit OS = Homo sapiens OX = 9606 GN = RPA1 PE = 1 SV = 2
Q8IYB3-2	SRRM1_HUMAN	62.27	4	6.64 × 10^6^	3	3	5	102,126	Isoform 2 of Serine/arginine repetitive matrix protein 1 OS = Homo sapiens OX = 9606 GN = SRRM1
Q8IYB3	SRRM1_HUMAN	62.27	4	6.64 × 10^6^	3	3	5	102,335	Serine/arginine repetitive matrix protein 1 OS = Homo sapiens OX = 9606 GN = SRRM1 PE = 1 SV = 2

**Table 5 pharmaceuticals-17-01435-t005:** Modes of adding aptamers to cell cultures.

Combination Name	Added aptamers			
	Day 1	Day 2	Day 3	Day 4
T	bi-(AID-1-T)	—	—	Start of observation
TT	bi-(AID-1-T)	bi-(AID-1-T)	—	
TC	bi-(AID-1-T)	bi-(AID-1-C)	—	
TTT	bi-(AID-1-T)	bi-(AID-1-T)	bi-(AID-1-T)	
TCT	bi-(AID-1-T)	bi-(AID-1-C)	bi-(AID-1-T)	
T-T	bi-(AID-1-T)	—	bi-(AID-1-T)	
T-C	bi-(AID-1-T)	—	bi-(AID-1-C)	
T + C	bi-(AID-1-T) bi-(AID-1-C)	—	—	
C	bi-(AID-1-C)	—	—	
CC	bi-(AID-1-C)	bi-(AID-1-C)	—	
CT	bi-(AID-1-C)	bi-(AID-1-T)	—	
CCC	bi-(AID-1-C)	bi-(AID-1-C)	bi-(AID-1-C)	
CTC	bi-(AID-1-C)	bi-(AID-1-T)	bi-(AID-1-C)	
C-C	bi-(AID-1-C)	—	bi-(AID-1-C)	
C-T	bi-(AID-1-C)	—	bi-(AID-1-T)	

**Table 6 pharmaceuticals-17-01435-t006:** List of primary human glioma cell cultures obtained from patient postoperative material with a brief description of tumors.

Culture Name	Grade	IDH1	Location
Sus\fP2	IV	IDH1 wild type	Left temporal lobe
BU307	IV	IDH1 wild type	Left temporal lobe
G-11	III–IV	IDH1 wild type	Right frontal–parietal–insular region.
G-01	IV	IDH1 wild type	Left frontal lobe
G-22	IV	IDH1 wild type	Left frontal lobe

## Data Availability

The data presented in this study are available on request from the corresponding author. The data are not publicly available due to the reason that some of the generated data are not published.

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
