# Peer review of "Unite and Conquer: Association of Two G-Quadruplex Aptamers Provides Antiproliferative and Antimigration Activity for Cells from High-Grade Glioma Patients"

_pharmaceuticals, 2024, doi:10.3390/ph17111435_

Round 1

Reviewer 1 Report

Comments and Suggestions for Authors
  1. Tables 1 and 3: These tables require more detailed explanations. Specifically, clarify which peptides were obtained from the 110-120 kDa band and which ones came from the 33-36 kDa band. Also, there's no mention of Table 2—please clarify its absence or include it if relevant.

  2. Figure 2: This figure needs much more clarification. Explain what the different colored lines represent in detail.

  3. FAM-tagging efficiency: The authors must demonstrate that the tagging efficiency with FAM was similar for both G-quadruplex structures. Without this, concluding that there was a differential intracellular accumulation is not fully justified.

  4. Dynasore and G-quadruplex endocytosis: To conclude a differential effect of Dynasore on the endocytosis of G-quadruplexes, the authors should include a quantification of numerous cells across at least three independent experiments. Statistical analysis should also be performed. Similarly, this analysis is needed for the results in Figure 5.

  5. Table 5: There are words in Russian (presumably), which should be corrected or clarified.

Minor:

  1. Introduction: Consider deleting the first paragraph of the introduction, as it contains basic and obvious information that would be redundant for readers of a specialist journal.

  2. Figure 1: It might be useful to add an experimental scheme (as panel A) to Figure 1 to clarify the interpretation of the results. Additionally, indicate how many times the experiment was replicated.

Comments on the Quality of English Language

There were some mistakes observed. Please check all the document.

Author Response

Dear reviewer,

I thank you for reading my article. I am sure that by correcting the aspects you have pointed out, I will make our article better. I have attentively read all your comments and suggestions and detailed answers are given in the attached file

Reviewer 2 Report

Comments and Suggestions for Authors

The manuscript describes the combined use of two aptamers in the treatment of gliomas with a special attention to deliver antiproliferative and antimigration activity on cells derived from patients. To this end, authors select two aptamers that were described to be active as antiproliferative agents in human glioblastoma cells and found that these aptamers bind PARP1 protein and others. They also describe the mechanisms of internalization of these aptamers. Then they study several combinations of the two drugs in on cells derived from patients finding a few combination that successfully inhibit proliferation as well as migration of tumoral cells. The results are rationalized by analysis of the expression of several genes involved in proliferation and migration of cells such as CD44, Nestin, PARP1, L1CAM, Sox2, p53, Caveolin1, and c-myc. The manuscript is interesting and it can be published after minor changes.

In Table 5 authors leave the text regarding the location of the tumors in Russian. This text that describe the position of the tumor (frontal, temporal, right or left) should be translated to English.   

Page 2, line 89: it says “his crypto-aptamer”, it should be  “This crypto-aptamer”

Page 8, line 243. It says “t was observed” it should be  “It was observed.

Author Response

Dear reviewer,

Thank you very much for taking the time to review this manuscript. I am grateful for your response and deeply appreciate your suggestions for improving the article.  We will definitely correct your remarks to the manuscript.

Reviewer 3 Report

Comments and Suggestions for Authors

The current manuscript by Pavlova and co-workers describes studies on various combinations of two G4-oligonucleotide aptamers [bi-(AID-1-T) and bi-(AID-1-C)] to identify additive or synergistic antiproliferative or antimigratory activity in glioma advanced-grade cells. The study specifically includes: (1) Identification of potential protein targets for both aptamers, which were not previously known, by means of cell lysis-isolation of cellular proteins-pull down of targets using the aptamers-PAGE-mass characterization; (2) Assessment of affinity of one of these proteins (PARP1, of interest to anticancer research) for one of the aptamers, by means of biolayer interferometry; (3) Study of cellular permeation for each of the aptamers over time, by means of fluorescence microscopy, and co-localization study with early endosomes- and lysosomes-targeted dyes; (4) Study of expression levels of several protein markers (CD44, nestin, PARP1, L1CAM, Sox2, p53, caveolin, c-Myc) in response to the aptamers and their combinations; (5) Proliferation inhibition (MTS assay) and (6) migration inhibition (Transwell) studies, in conjunction to cell treatment with various aptamer combinations. 

While the idea of employing combinations of aptamers and/or aptamer+drug to detect additive or synergistic effects is not new, and has been shown by others on numerous occasions to be beneficial, combinations of these two aptamers have not been studied before, and the findings are promising. However, there are numerous reasonable gaps in this study, and major revisions (and perhaps additional studies) would be needed to render this manuscript acceptable for publication:

(1) Several protein targets are identified for each of these aptamers (broad selectivity), and a stronger case needs to be made why only one of them was chosen for further investigation, but most importantly how may this protein target be connected with the observed cellular effects, i.e., is the observed inhibition of proliferations and migration PARP1-mediated, for example? This has not been convincingly discussed.

(2) The two aptamers have quite similar sequence and yet very different mechanisms for cellular uptake are suggested by the results: dynamin-mediated endocytosis for bi-(AID-1-T), but a different -albeit undetermined- path for bi-(AID-1-C). No rationale is given for this dual behaviour, despite the extensive sequence similarity. Is it a 3D-related issue? How do the topologies of the two aptamers differ in a way that could affect cellular entry?

(3) A slow and rather negligible cellular uptake of bi-(AID-1-C) is suggested by the fluorescence microscopy, which does not seem to increase significantly after the first 24 h of incubation. And yet, the combination of aptamers T and C appears to be one of the most efficient for the cellular index of glioma cell cultures, showing an increase in percentage during the 4-7 days window (the 0-4 days window is not included in the study). How are the two facts reconciled? No attempt is made to explain this in relation to the cellular uptake of the two aptamers, and especially the problematic cell entry of bi-(AID-1-C).

(4) In line 259, the authors mention that: "there was some overlap between aptamers and early endosomes". However, the images are not really suggestive of that, as there are almost no "yellow" co-localization areas in the overlay, suggesting that the aptamer location may be different from that of the early endosomes.

(5) In Figure 7 and related text, please clearly define what is the control used and what does the % in the graphs' vertical axis represent. It is not very clear to a non-specialist reader.

(6) The results of Figure 7 suggest that a triple combination may in fact be worse that the double combination, and in certain cases worse than one single aptamer. Why is that? No rationale is provided.

(7) The Discussion section is too extensive and much of the information included there is general or common knowledge. I find the inclusion of such information here quite disruptive. Much of it should be shifted to the Introduction section, and only specific information directly related to the aptamer system at hand should be retained in the Discussion. The Discussion should be more concise, linking research findings to reasonable conclusions. Repetitions should be avoided, as well as claims not supported by the results. The connection of the studied protein markers to both proliferation and migration should be better established in the Discussion, if any.

(8) The Introduction is somewhat general. It can be enriched with more information on aptamers and their applications, some of which is currently found in the Discussion section. The Introduction should also provide statistics in regard to mortality rates or success rates of aptamers when used as treatments in any mentioned examples.

(9) In section 2.2, I would find it helpful if the authors can include the determined binding affinity values in a Table rather than mentioning the in the text.

(10) The English throughout the text should be carefully checked by an avid speaker, as corrections of grammar mistakes and typos are needed at several points.

Comments on the Quality of English Language

See point 10 above.

Author Response

Dear Reviewer,

Thank you for taking the time to read my article so thoroughly. I sincerely appreciate your constructive feedback. I am confident that addressing the aspects you highlighted will enhance the quality of our work significantly. I have carefully considered all of your comments and suggestions, and I have provided a detailed response to each point below.

(1) Several protein targets are identified for each of these aptamers (broad selectivity), and a stronger case needs to be made why only one of them was chosen for further investigation, but most importantly how may this protein target be connected with the observed cellular effects, i.e., is the observed inhibition of proliferations and migration PARP1-mediated, for example? This has not been convincingly discussed.

For each aptamer, the protein with the most explicit peak in MS data has been chosen. The proteins are briefly described in the Results section (lines 226-237, 345-358), and their possible roles in tumorigenesis were also described in the Discussion section (lines 504-530). The exact roles of these proteins in observed cellular effects for glioblastoma cells: migration and proliferation, is not known yet. These questions could be addressed after discovery only, and usually are topics of separate large research.

(2) The two aptamers have quite similar sequence and yet very different mechanisms for cellular uptake are suggested by the results: dynamin-mediated endocytosis for bi-(AID-1-T), but a different -albeit undetermined- path for bi-(AID-1-C). No rationale is given for this dual behaviour, despite the extensive sequence similarity. Is it a 3D-related issue? How do the topologies of the two aptamers differ in a way that could affect cellular entry?

A sequence similarity of these aptamers is limited by a similarity of GQ topologies. The loops are different and therefore endocytosis mechanism could be different due to an involvement of number of different transporter proteins. This issue requires further detailed study, which we are planning to do.

(3) A slow and rather negligible cellular uptake of bi-(AID-1-C) is suggested by the fluorescence microscopy, which does not seem to increase significantly after the first 24 h of incubation. And yet, the combination of aptamers T and C appears to be one of the most efficient for the cellular index of glioma cell cultures, showing an increase in percentage during the 4-7 days window (the 0-4 days window is not included in the study). How are the two facts reconciled? No attempt is made to explain this in relation to the cellular uptake of the two aptamers, and especially the problematic cell entry of bi-(AID-1-C).

We are also curious with that result. Since FAM-labeled aptamers were made by automatic synthesis, and not by chemical addition of FAM, in both cases bi-(AID-1-T) and FAM bi-(AID-1-C) have equal amounts of identically labeled aptamers (purity data sheets can be provided upon request). The reason could be differences in aptamer penetration into tumor cells, and accumulation rates. Here, we only have noticed that even a small uptake of bi-(AID-1-C) into cells after bi-(AID-1-T) has significantly reduced a migration of glioma tumor cells. The details of this effect remain to be further studied.

(4) In line 259, the authors mention that: "there was some overlap between aptamers and early endosomes". However, the images are not really suggestive of that, as there are almost no "yellow" co-localization areas in the overlay, suggesting that the aptamer location may be different from that of the early endosomes.

Arrows have been added to pinpoint overlaps

(5) In Figure 7 and related text, please clearly define what is the control used and what does the % in the graphs' vertical axis represent. It is not very clear to a non-specialist reader.

Requested explanations have been added into the Figure Legend

(6) The results of Figure 7 suggest that a triple combination may in fact be worse that the double combination, and in certain cases worse than one single aptamer. Why is that? No rationale is provided.

An impact of different aptamers on cells depends on various steps which are poorly studied by now, since the phenomenon has just been discovered. Aptamers penetrate into a cell by different pathways, using different transporter proteins; after entering the endosomes, complexes are dissociated, and aptamers are released in different ways, bind to different proteins, and so on, which produce different effects. This scenario requires a step-by-step study.

(7) The Discussion section is too extensive and much of the information included there is general or common knowledge. I find the inclusion of such information here quite disruptive. Much of it should be shifted to the Introduction section, and only specific information directly related to the aptamer system at hand should be retained in the Discussion. The Discussion should be more concise, linking research findings to reasonable conclusions. Repetitions should be avoided, as well as claims not supported by the results. The connection of the studied protein markers to both proliferation and migration should be better established in the Discussion, if any.

Yes, we agree, that the Discussion section is kind of too extensive. We have followed your advice to shift some part to the Introduction section.

(8) The Introduction is somewhat general. It can be enriched with more information on aptamers and their applications, some of which is currently found in the Discussion section. The Introduction should also provide statistics in regard to mortality rates or success rates of aptamers when used as treatments in any mentioned examples.

We do appreciate your advice. The Introduction section have been supplemented with parts from the Discussion section. Yet, there are no data on clinical results in question. Should you recommend to include into manuscript some data on the therapeutic oligonucleotides trails from the site <ClinicalTrials.gov>?

(9) In section 2.2, I would find it helpful if the authors can include the determined binding affinity values in a Table rather than mentioning the in the text.

The Table has been added: Table 2, line 204

(10) The English throughout the text should be carefully checked by an avid speaker, as corrections of grammar mistakes and typos are needed at several points.

The text has been given to the native speaker, and recommended edits have been made

Round 2

Reviewer 1 Report

Comments and Suggestions for Authors

The concerns were answered. 

Author Response

I thank the Reviewer for the response. It was very important to me.

Reviewer 3 Report

Comments and Suggestions for Authors

After the changes done by the authors in response to the reviewer's comment, I now find the manuscript fit for publication. Concerns that still remain are:

- On point #3, would the authors consider the possibility that the aptamer that does not efficiently enter cells may be doing something extracellularly to affect the cellular index? Maybe add a sentence to cover this possibility in the manuscript. 

- On point #8, yes, it'd be useful to include some representative clinical data, just enough to make the point.

Author Response

I thank the Reviewer for the response.

It could not be excluded that bi-(AID-1-C) is able to exert some extracellular impact, and this requires further thorough study. This note was included in the manuscript.

Oligonucleotide therapeutics medical trials are at the very juvenile stage. On October 15, 2024, in the DB ClinicalTrials.gov there are only 227 notes for ‘oligonucleotides plus therapeutics’, 36 notes with the results. The most interesting case is Duchenne Muscular Dystrophy.